# Plasmonic Phenomena in Membrane Distillation

**DOI:** 10.3390/membranes13030254

**Published:** 2023-02-21

**Authors:** Francesca Alessandro, Francesca Macedonio, Enrico Drioli

**Affiliations:** Institute on Membrane Technology, National Research Council of Italy (CNR-ITM), Via P. Bucci 17/C, 87036 Rende, Italy

**Keywords:** temperature polarization, plasmonic nanostructures, thermoplasmonic membrane distillation

## Abstract

Water scarcity raises important concerns with respect to human sustainability and the preservation of important ecosystem functions. To satisfy water requirements, seawater desalination represents one of the most sustainable solutions. In recent decades, membrane distillation has emerged as a promising thermal desalination process that may help to overcome the drawbacks of traditional desalination processes. Nevertheless, in membrane distillation, the temperature at the feed membrane interface is significantly lower than that of the bulk feed water, due to the latent heat flux associated with water evaporation. This phenomenon, known as temperature polarization, in membrane distillation is a crucial issue that could be responsible for a decay of about 50% in the initial transmembrane water flux. The use of plasmonic nanostructures, acting as thermal hotspots in the conventional membranes, may improve the performance of membrane distillation units by reducing or eliminating the temperature polarization problem. Furthermore, an efficient conversion of light into heat offers new opportunities for the use of solar energy in membrane distillation. This work summarizes recent developments in the field of plasmonic-enhanced solar evaporation with a particular focus on solar-driven membrane distillation applications and its potential prospects.

## 1. Introduction

Water scarcity is one of the major problems facing many societies and the world in the 21st century due to the rapid growth of the population as well as industrial activities [1,2,3]. According to a World Health Organization report, as soon as 2025 [4], half of the world’s population could live in areas prone to water scarcity. This problem is particularly widespread in less developed countries, where 38% of healthcare institutions have water supply problems [5]. On our planet, the ocean holds about 97% of the Earth’s water; the remaining 3% is available as fresh water, although most of this exists in the form of glaciers and ice caps (~2%), leaving less than 1% of the fresh water available for drinking.

In this scenario, desalination is necessary to produce potable water by removing dissolved salts and minerals from saline water [6]. Current commercial utilized desalination technologies include solar still distillation (SD), multistage flashing (MSF), multi-effect distillation (MED), reverse osmosis (RO), and electrodialysis (ED). RO processes are widely used and represent 60% of desalination plants worldwide due to their lower energy requirements than other technologies, such as thermal desalination [7]. Despite this undeniable advantage, RO presents important limitations, including the disposal of the produced brine and a limited potential to further reduce energy consumption [8,9]. To overcome these issues, several techniques have been developed, including membrane distillation (MD), which is recognized as a promising and innovative technology [10].

Membrane distillation is a thermally driven membrane process in which a heated aqueous solution is placed in contact with one side (feed/retentate side) of a hydrophobic membrane that inhibits the mass transfer of the liquid and creates a vapor–liquid interface at the pore entrance. Volatile compounds evaporate, diffuse and/or convect across the membrane and then are condensed on the opposite side of the membrane (distillate/permeate side) [11,12,13,14]. The driving force of the separation process is the partial pressure difference between the two sides of the membrane. According to the nature of the distillate side of the membrane, MD systems are classified into four basic configurations [15]: direct contact membrane distillation (DCMD), air gap membrane distillation (AGMD), sweeping gap membrane distillation (SGMD), and vacuum membrane distillation (VMD). The advantages and disadvantages of each configuration are summarized in Table 1.

In recent years, new configurations with improved energy efficiency and permeation flow have been developed, including material gap MD, permeate gap MD, multi-stage MD, multi-effect MD, vacuum multi-effect MD, and hollow fiber multi-effect MD [16,17,18].

Compared to pressure-driven systems, MD processes offer some important advantages such as lower operating temperatures, lower operating pressure, high rejection, and performance not limited by high osmotic pressure or concentration polarization. These properties make MD particularly attractive for the treatment of wastewater and the desalination of water [11,19]. Furthermore, the possibility of integrating MD and crystallization could pave the way for a promising hybrid separation process capable of achieving high water recovery, a suitable controlled saturation ratio, high crystallization, low induction time, and low fouling potential [15,20].

However, the traditional MD process is characterized by considerable drawbacks, including high energy input for heating the bulk feed water, heat loss during the feed transportation from heating units to membrane modules, and the need for large central pumping systems [21]. In addition, its large-scale application is also limited by its low thermal efficiency due to the temperature polarization (TP). This phenomenon is related to latent heat and conductive heat transfer at the MD membrane interface [22,23]. The feed temperature at the membrane surface $\left(T_{fm}\right)$ is lower than the bulk feed water temperature $\left(T_{f}\right)$ due to thermal conductivity and water vaporization. Likewise, the temperature at the distillate–membrane interface $\left(T_{dm}\right)$ is higher than the temperature of the bulk distillate $\left(T_{d}\right)$. Therefore, TP significantly reduces the temperature difference at the membrane interface with respect to the theoretical driving force across the bulks (Figure 1a).

Several solutions have been suggested to mitigate the negative effects of TP [23]. Some studies reported the use of frame-like turbulence promoters (e.g., feed spacers) [24] or modified feed channels (e.g., corrugated channels) [25] to alleviate TP and thus control MD performance and improve distillation efficiency. Nevertheless, the use of these approaches leads to an increased demand for energy [26].

Because TP is an inherent loss process that cannot be completely removed, it is desirable to also explore other strategies that could alleviate this phenomenon and improve MD performance.

In particular, thermoplasmonics [27] could offer new opportunities for MD, thanks to the possibility of harnessing the kinetic energy of light to generate and control a large amount of heat at the nanoscale. Over the past few years, the use of photothermal materials acting as nanosources of heat in the MD membranes has been proposed for TP mitigation, contributing to the development of a new MD configuration, called photothermal membrane distillation (PMD) [28].

In the literature, various photothermal materials developed for solar steam generation (SSG) systems [29] have been employed to realize PMD membranes, including plasmonic materials [30,31], carbon-based nanomaterials [32,33], inorganic semiconductors [34], and polymers [35]. Most notably, plasmonic nanostructures (Au, Ag, MXene, TiN, etc.) have intensive light absorption and strong photothermal conversion capacities [36].

The plasmonic heating mechanism is based on a nonradiative conversion of light to heat by plasmonic nanostructures [27]. As shown in Figure 2a, plasmonic materials exhibit localized surface plasmon resonance (LSPR) when the frequency of the incident light matches the resonance frequency of the free electrons. The excitation of electrons is triggered by the match between the frequencies. Following the photoexcitation, the surface plasmons can behave as strongly damped oscillators and decay through various channels of interaction [37] (electron-to-photon, electron-to-electron, and electron-to-phonon) that allow the dissipation of the plasmonic energy in thermal energy [38] (Figure 2b).

Specifically, in the timescale ranging from 1 to 100 fs after Landau damping, the electron–hole pairs can decay either by emission of photon or by carrier multiplication due to the electron–electron interactions [37,39]. The hot carriers will redistribute their energy via electron–electron scattering processes in a few hundred femtoseconds. Lastly, the heat will dissipate by thermal conduction in the environment surrounding the nanostructure on a time scale between 100 ps and 10 ns.

For a single metallic NP surrounded by a homogeneous medium, the increase in temperature due to the light irradiation is given by the following equation [40]:$$\Delta T_{NP}=\frac{\sigma_{abs}I}{4\pi R\kappa_{m}}$$
where $\sigma_{abs}$ is the absorption cross-section, I is the irradiance of the illumination, R is the radius of the NP, and $\kappa_{m}$ is the thermal conductivity of the surrounding medium.

In general, the contribution of a sequence of metallic plasmonic nanoparticles (NPs) surrounded by a homogeneous medium can be obtained by solving the following heat transfer equation:$$\rho \left(r\right)C_{p}\frac{\partial T\left(r\right)}{\partial t}=\nabla \times \left[k\left(r\right)\nabla T\left(r\right)\right]+Q\left(r\right)$$
where ρ, $C_{p}$, T, and k Q are the density, specific heat capacity, temperature, thermal conductivity, and local heat intensity, respectively, expressed as a function of coordinate r [41].

As a result, it is difficult to assess the impacts of nanosources of heat on the membrane temperature.

Recently, Elmaghraoui et al. [42] modified the previous model by the addition of a term associated with heat losses into the heat transfer equation. This new approach includes the cooling effects related to vaporization at the membrane surface. According to this, the membrane steady-state temperature $\left({\Delta T}_{m}^{ss}\right)$ is reported in the following equation:$$\Delta T_{m}^{ss}=-\frac{{ℒ}_{k\tau }}{k}\left\{\left(1+\frac{R_{m}}{\sqrt{\beta \tau }}\right)e^{\frac{-R_{m}}{\sqrt{\beta \tau }}}\right\}+\frac{ℒ\beta \tau }{k}$$
where ℒ is the effectively continuous heat production due to absorbed energy on NPs that depends on the excitation energy, the intensity, the NP concentration, and their size. $R_{m}$ is the radium of a sphere with a volume of measured suspensions, and τ is a cooling time constant. The introduction of the cooling term in the theoretical model provides an even more accurate description of the experimental results.

Both models suggest that the membrane temperature increases with the number density of the NPs used as filler. Therefore, the use of nanosources of heat in MD processes would lead to a temperature increase in a large area (Figure 1b), resulting in an improvement in mass transfer.

However, to date, the practical application of photothermal membranes is hindered by many obstacles, including membrane wetting, fouling, scaling degradation, and the need to redesign the MD membrane module.

This review outlines recent progress in the integration of plasmonic nanostructures in porous membranes for MD applications, and the role of thermoplasmonics in solar technology. In particular, recent advances are reviewed for photothermally assisted evaporation and PMD. Finally, prospects and challenges are also discussed.

## 2. Plasmon-Enhanced Solar Evaporation

### 2.1. Photothermal Conversion

In solar evaporation systems, photothermal material is one of the key components that determine evaporation efficiency. An ideal photothermal material should meet several important requirements, such as highly efficient optical absorption, high light-to-heat conversion, proper thermal management, and ease of scale-up. The amount of light energy which can be converted into thermal energy is related to the capacity of the photothermal material to absorb solar radiation [43]. Specifically, the solar absorbance (α) is determined by the ratio of the total absorbed solar radiation to the incident radiation:$$\alpha =\frac{\int_{\lambda_{min}}^{\lambda_{max}}I\left(\theta ,\lambda \right)\left[1-R\left(\theta ,\lambda \right)\right]d\lambda }{\int_{\lambda_{min}}^{\lambda_{max}}I\left(\lambda \right)d\lambda }$$
where $\lambda_{min}$ and $\lambda_{max}$ are 300 and 2500 nm, respectively, θ is the incident angle of light, I(λ) is the wavelength-dependent solar spectral irradiance, and R(θ,λ) is the total reflectivity [44].

### 2.2. Solar Evaporation Systems

In general, the position of the photothermal material in the solution determines the type of solar energy evaporation system. In volumetric systems (Figure 3a), the photothermal materials are dispersed in the bulk fluid to generate heat. These systems achieve a moderate efficiency (between 20% and 60%) under one sun [45,46] due to radiation, conduction, and convection heat losses. In bottom systems (Figure 3b), the solar absorbers are located at bottom of the vessel, and the vapor is produced in a separate place. The significant heat losses that occur reduce the evaporation efficiency of these systems to as low as 30–40% [47]. Novel interfacial solar evaporation systems (Figure 3c) where photothermal conversion and vapor generation are located at the water–air interface have been designed. Due to the lower temperature of the absorber, radiation and convection heat losses are reduced in these systems. In addition, since the thermal energy is confined on the absorber surface, the energy losses due to volumetric heating are reduced. Because of this, interfacial systems can achieve solar-to-steam conversion efficiencies of over 90% under one sun [48]. However, the high contact area between photothermal material and water generates high conduction heat losses. To mitigate heat loss, systems in which the photothermal material is isolated from bulk fluid have been designed (Figure 3d). In this operation mode, water is carried to the photothermal material using a water channel or enclosed in a sponge [49].

### 2.3. Plasmonic Solar Absorbs in Solar Evaporation Systems

As mentioned before, nanoscale heating is obtained due to the surface plasmon resonance of metal nanostructures which under solar irradiation can generate heat nanosources due to strong light absorption at the plasmonic resonance wavelength range. The plasmon energy transformed into heat increases the temperature of the surrounding fluid [50]. Due to their chemical stability, tunable size and shape, and optical properties, gold (Au) and silver (Ag) NPs are commonly used as photothermal materials. Their LSPR peaks fall in the infrared region of the visible spectrum. In the same way as other metal NPs, noble metal NPs have strong plasmon resonance absorption at specific spectral wavelengths. A widening of the resonance band can be achieved by modifying the size and shape of the nanostructures [51] or by creating a plasmonic coupling between two plasmonic NPs [52].

The first plasmon-assisted solar evaporation investigation was reported by Halas’s group [45], which dispersed SiO_2_/Au NPs in water to improve the evaporation process (Figure 4a). Thermodynamic analysis revealed a solar-to-steam generation efficiency of only 24%; however, this pioneering work has inspired researchers to investigate the potential of noble metals in solar energy applications.

To reduce the thermal energy loss, Bae et al. [53] realized thin-film black gold membranes which have multiscale structures of varying metallic nanoscale gaps as well as microscale funnel structures (Figure 4b). The heat is located within the film of a few micrometers and a continuous supply of water is guaranteed through the micropores. This Au film recorded 57% conversion efficiency under light illumination of $20\;\mathrm{kW}\;m^{-2}$. The size and morphology of nanoparticles are two crucial factors for photothermal conversion [54]. Guo et al. [55] used Au NPs with different sizes to investigate the diameter effect on the photothermal conversion efficiency. The experimental results revealed that the photothermal conversion efficiency was enhanced (by about 2% under 1 sun) by increasing the Au NPs’ diameters (from 3 to 40 nm) in line with the theoretical calculations. In addition to size, the variation of the NP shape can also be exploited to tune its localized surface plasmon (LSP) responses, as the nanostructure shape strongly influences the light–matter interaction.

A classic example is represented by the anisotropic Au nanorods [56,57], that unlike spherical Au NPs, exhibit two specific LSPR absorption bands. Au nanoroads, compared to spheres with the same volume, could be more efficient plasmonic absorbers and photothermal converters due to their larger surface-to-volume ratio. The efficiency could be increased by ∼60% from spherical to nanorod [58].

Recently, Yougbare et al. [59] found that Au NPs, in the form of nanobipyramids, exhibit a higher photothermal performance compared to Au nanorods under light irradiation. Through the shape manipulation, Au nanocages (AuNCs) were prepared with an absorption peak covering the visible and near-infrared (NIR) regions and incorporated into electrospun nanofibers of a polyvinylidene fluoride (PVDF) membrane [60]. With the increase in the amount of AuNCs, the solar evaporation efficiency exhibits a notable improvement from 67.0% to 79.8% when using AuNCs at weight percentages of 0.05% and 0.10%, respectively [60]. However, the high costs of Au NPs hinder their solar-driven applications.

In comparison with Au NPs, Ag NPs are cheaper and exhibit desirable properties, such as higher conductivity, higher chemical stability, and higher plasmonic resonance response. Specifically, photothermal effects in Ag NPs are 10 times higher than in Au NPs [61], which is particularly interesting for surface plasmon applications. An efficient solar vapor generator was realized, coating Ag NPs on a diatomite surface (Ag/D) [62]. Ag/D, filtered onto filter papers and wrapped around polystyrene foam, showed an evaporation rate of ca. $1.39\;{\mathrm{kg}\;m}^{-2}{\;h}^{-1}$ and efficiency of ca. 92% under 1 sun illumination [62]. The high performance of the system was attributed to the combination of localized plasmonic effects of Ag NPs and the confinement effect in micrometer-sized diatomite. Black Ag nanostructures with widely distributed interparticle distances have been prepared in rod-shaped tubular spaces [63]. This approach has led to a strong random plasmonic coupling liable for broadband absorption. Solar steam efficiency of about 95% was recorded when the Ag nanostructures were deposited into thin films and employed for solar evaporation investigations. However, the toxicity potential of the Ag NPs when they are employed in a solar evaporation system should be fully assessed. Solar photothermal conversion performance can also be optimized by mixing different NPs to match the solar spectral intensity [64]. Zhu et al. [65] prepared Ag-Au/ZIF-8-derived nitrogen-doped graphitic polyhedron (Ag-Au/ZNG) nanofluids by the impregnation-reduction method. Plasmonic Ag-Au nanoparticles improved the photothermal conversion efficiency, which was enhanced by up to 74.35% for Ag-Au ZNG nanofluids compared with 72.41% and 70.35% for Au/ZNGs and Ag/ZNGs, respectively [65].

Less expensive alternatives to Au and Ag are copper (Cu) and aluminum (Al).

Due to its strong Vis-NIR absorption and omnidirectional light absorption properties, Cu nanoparticles show LSPR peaks at longer wavelengths compared to Au NPs and Ag NPs [66]. Nevertheless, Cu NPs exhibit less resistance to corrosion in aqueous solutions and oxidation in air than Au and Ag NPs. To overcome these obstacles, Cu NP films have been covered with thin protective coatings or treated with corrosion inhibitors [67,68]. In recent years, Zhang et al. [69] developed a monolayer of monodisperse microparticles with nanoengineered surface compositions and structures self-floating on the water surface. The microparticles contain a plasmonic Cu component dispersed on a TiO_2_ matrix which absorbs high broadband solar energy and converts it into thermal energy, a carbonaceous component to improve light absorption, and a silica component to reduce heat losses. Under the illumination of 1 sun, a photothermal efficiency of 92.2% was recorded [69]. Likewise, Cu NPs prepared through a substitution reaction by Li et al. [70] showed a high photothermal conversion efficiency of 93%. Moreover, by coating the Cu NPs on a cellulose membrane, a solar steam generation efficiency of up to 73% was acquired at 1 sun irradiation.

Al exhibits strong plasmonic excitation in the UV region. In order to achieve efficient solar-to-heat conversion, it is crucial to expand its spectral range toward IR. As a result of the intrinsic oxidation of Al, the LSPR absorption can be red-shifted and thus broadened [71]. In this context, Al NPs incorporated into a 3D nanopore anodized aluminum oxide membrane (AAM) enabled an energy conversion rate of ca. 90% and a broadband solar absorption of >96% [72].

Plasmonic-like properties of tellurium (Te) in the solar radiation region were reported for the first time by Yang et al. [73]. Over 85% of the solar energy can be absorbed by using Te NPs as absorbers [73]. However, Te is a water pollutant [74]; in order to overtake this drawback, Te NPs have been covered with a selenium (Se) layer that enhances Te biocompatibility and also limits its oxidation [75].

MXenes are two-dimensional (2D) transition metal carbides nitrides, and/or carbonitrides. Ti_3_C_2_ [76,77,78], one of the most investigated MXene materials, has shown photothermal efficiency of 100% in the visible range [79]. The application of hydrophobic Ti_3_C_2_ membrane in desalination exhibited a conversion efficiency of 71% under one sun [80].

Plasmonic response in the Vis-NIR range was also exhibited by titanium nitride (TiN) and zirconium nitride (ZrN) [81]. For instance, TiN NPs exhibited an average absorbance of more than 95% [82].

In addition, as a result of their exceptional light absorption and photothermal conversion properties, plasmonic semiconductors have recently attracted much attention. Their LSPR response can be controlled by self-doping or heterovalent doping [83,84]. WO_3−x_ and TiO_1.67_ showed effective LSPR absorption in near- and mid-infrared wavelengths [85,86]. Photothermal conversion properties have been reported in other semiconductor NPs such as rubidium tungsten bronze (RbxWO_3_) [87] and copper-deficient copper chalcogenides, such as Cu_2_−xSe (X = Al, Ga, In) [88].

The performances of some plasmonic NPs in SSG are summarized in Table 2.

## 3. Photothermal Membrane Distillation

### 3.1. PMD Process

Similar to conventional membrane distillation, in the PMD system, the feed solution is separated from the cold distillate by a hydrophobic membrane. The hydrophobic nature of the membrane prevents the intrusion of aqueous solution in the membrane pores, creating a vapor–liquid interface at each pore entrance [28]. Then, the vapor passes across the membrane pores and is condensed on the distilled side of the membrane.

However, in the PMD process, plasmonic photothermal materials (dispersed into/onto the membrane) collect solar energy at the feed–membrane interface, providing localized heating at the evaporation surface and thus reducing the external energy input required for heating the feed [28,35,89].

In PMD, the heat transfer processes include the conduction and convection terms, as in conventional MD, and a radiation heat loss contribution due to the fact that all objects absorbed and remitted electromagnetic radiation. The PMD heat transfer equations are given below [43]:-Conduction through a medium:
$$q_{cond}=\frac{k}{\delta }\times \Delta T$$
where k is the thermal conductivity coefficient, δ the thickness of the medium, and ΔT is a temperature difference.
-Convection through a moving fluid:
$$q_{conv}=h\;\left(T_{f}-T_{m}\right)$$
where *h* is the convective heat transfer coefficient $\left({W\;m}^{-2}k^{-1}\right)$, $T_{m}$ is the temperature of the membrane surface, and $T_{f}$ is the temperature of the surrounding fluid.
-Electromagnetic radiation:
$$q_{rad}=\varepsilon \sigma \;\left(T_{m}^{4}-T_{f}^{4}\right)$$
where ε is the emissivity of the membrane in the fluid, and σ is the Stefan–Boltzmann constant $\left(5.67\times 10^{-8}{\;W\;m}^{-2}{\;k}^{-4}\right)$.

In comparison with the conventional MD, the PMD system exhibits important benefits, such as the possibility to save part of the energy used to heat up the feed solutions, to reduce the energy consumption of pumping feed solutions, and to mitigate the heat loss due the transport of hot feed solutions [21].

Another key parameter in PMD is the solar conversion efficiency (η), which can be calculated in accordance with the following equation:$$\eta =J\frac{\;\left(h_{V}+Q\right)}{I}\;\left(25\right)$$
where J is the permeate flux across the membrane, $h_{V}$ is the water evaporation enthalpy, *Q* is the sensible heat required to increase the initial temperature of the system to the evaporation temperature, and *I* is the irradiance of illumination.

### 3.2. PMD Systems

PMD can operate in different configurations, listed below:DCMD-based PMD: In this system, the photothermal membrane is illuminated by the radiation at the feed side, whereas the permeate side is distillate water. An advantage of this system is that the vapor can be condensed inside the module. In contrast, the main disadvantages came from the high conductive heat losses, which reduce the temperature difference across the membrane surface [90,91,92].AGMD-based PMD: In this process, only the feed solution is in direct contact with the membrane surface, and an air gap is introduced between the membrane and condensing surface. The presence of the air gap reduces heat loss due to the conduction through the membrane and mitigates the membrane-wetting tendency. Conversely, the permeate fluxes are generally low as a result of the additional mass transfer resistance from the air gap [93].SGMD-based PMD: In this operation, an inert gas sweeps vapor molecules from the membrane permeate side into an external condenser (placed outside the membrane module) where vapor condensing occurs. A reduced mass transfer resistance is responsible for the higher permeate flux produced by photothermal SGMD in comparison to photothermal AGMD. Nonetheless, since the permeate must be collected outside the module and the system design is more complicated, photothermal SGMD is the least used configuration [94].VMD-based PMD: In this configuration, the photothermal membrane is illuminated by the radiation at the feed side while a vacuum pressure is applied on the permeate side. Vapor condensing occurs in an external condenser through a vacuum pump. This process is able to mitigate the reduction in temperature difference caused by membrane thermal conduction and reach high transmembrane flux. However, due to the applied vacuum that creates a significant variation in pressure at the membrane interface, this configuration requires high adhesion stability of the loaded photothermal materials and high mechanical strength of the membrane. In addition, as a result of the complexity of this setup, heat recovery may be technically difficult [28].

As previously mentioned, PMD processes aim to improve the temperature gradient across the membrane and mitigate temperature polarization effects. For a high transmembrane flux, it is crucial to direct the thermal energy to the targeted area. Localized heating causes an increase in the feed temperature at the membrane surface, resulting in an increase in the temperature difference between the feed and the distillate sides and thus in the driving force of the process. This temperature difference could be further increased by reducing conductive heat loss from the feed side to the permeated side. Low thermal conductivity of the membrane material, together with high membrane porosity and thickness, are factors for minimizing the heat transfer between feed and permeate [95]. Moreover, if the photothermal layer is located on the top of the hydrophobic membrane, its high thermal conductivity can be exploited for the efficient heat transfer to the evaporation surface [96].

Heat transfer is also influenced by the feed flow rate in photothermal MD. Conversely to the conventional MD, where the permeate flow increases with the feed flow rate, the PMD system exhibits greater efficiency with a lower feed flow rate [21,89]. This allows a longer exposure time to light irradiation, causing the temperature gradient across the membrane to rise. The low flow rate enhances the energy efficiency of the PMD system, reducing the energy needed for water circulation. Therefore, these systems can be more attractive for off-grid areas.

The heat transfer equations for DCMD-based PMD, SGMD-based PMD, and AGMD-based PMD are reported in Table 3.

### 3.3. Module Design

The photothermal MD permeate flux can be also affected by the design of the membrane module (Figure 5). In conventional MD, the highest temperature is recorded at the feed inlet, causing the greatest transmembrane temperature difference. Conversely, because the feed temperature increases with the hot flux channel as a result of the longer heating time, in photothermal MD, the greatest temperature difference is observed near the feed outlet. Consequently, an increase in module size is expected to lead to an increase in permeate flux [89]. An appropriate membrane module design, especially the design of the feed channel, is also required in order to better control the thickness of the water layer on the membrane feed side [21]. A high depth of the water layer at the feed channel could indeed lead to a decrease in photothermal efficiency due to the occurrence of scattering and refraction phenomena [34]. Moreover, since the spiral and hollow fiber modules have a surface area not sufficient to receive enough light, it is preferable to use plate and frame membrane modules for photothermal MD applications [21].

### 3.4. Plasmonic Photothermal Materials in PMD Systems

Few studies have investigated and developed plasmonic materials for solar-driven MD applications (Table 4). In a groundbreaking study on photothermal VMD, Politano et al. [28] fabricated, via NIPS, a photothermal membrane by incorporating plasmonic Ag NPs into a PVDF polymer matrix. The photothermal plasmonic heat generated by Ag NPs increases the interface temperature under UV radiation, determining a clear improvement in the process. In fact, as a consequence of the thermoplasmonic effects induced by UV radiation, the authors reported a 11-fold higher distillate flux than the corresponding values for pristine PVDF membrane [86] (Figure 6).

Avci et al. [94] fabricated photothermal membranes by immobilizing Ag NPs on the top layer of the PVDF matrix. The membranes were fabricated by integrating a VIPS step before NIPS. The procedure allowed them to obtain photothermal membranes with adequate hydrophobicity, hence suitable for desalination operations carried out at high concentrations. In particular, the authors found that noble metal nanofillers led to an increase in transmembrane flux under UV radiation by about 10-fold with respect to unloaded PVDF membrane in a photothermal SGMD configuration. In order to collect electrochemical energy, the hypersaline retentate was fed to a reverse electrodialysis (RED) unit producing a maximum power density of up to $0.9{\;\mathrm{Wm}}_{\mathrm{MP}}^{-2}$ [94].

As nanotechnology progresses, new materials with multiple functionalities have been developed, and their potential practical uses have not yet completely been exploited. MXene is one such material, whose high photothermal efficiency depends on its plasmonic properties. Electron energy loss spectroscopy (EELS) experiments revealed that MXene exhibits both transversal and longitudinal surface plasmon modes ranging from visible up to 0.1 eV in the NIR region [97]. A reduction in the energy consumption per unit distillate volume by 12% has been observed when MXene nanostructures were incorporated in a PVDF membrane [89]. Additionally, the photothermal properties of MXene conferred outstanding anti-fouling properties to the membrane. However, the MXene coating imposed an additional mass resistance that lowered the flux by 13% [92].

Recently, TiN NPs, due to their broadband light absorption, high light-to-heat conversion efficiency, high chemical stability, and biocompatibility, have attracted considerable attention in plasmonic heating applications. TiN photothermal membranes were prepared using a TiN NP/PVA-doped solution, spray-coated onto a hydrophobic PVDF membrane surface [91]. As a result of the absorption of broadband light and the superior heat conversion properties of plasmonic TiN NPs, the TiN photothermal membrane showed a permeate flux of $1.01{\;L\;m}^{-2}{\;h}^{-1}$, with a solar efficiency of 66.7% under 1 sun in the photothermal DCMD process [91].

Using a two-step method (i.e., electrospinning coating followed by heat crosslinking), Zhang et al. [93] realized a dual-layer membrane with a TiN@PVA photothermal coating layer on a PVDF support layer. With 2 mm of stagnant feedwater on top of the aforementioned optimal membrane, a flux of $0.640\;{\mathrm{kg}\;m}^{-2}{\;h}^{-1}$ and a solar efficiency of 64.1% were achieved in photothermal AGMD operation. In addition, the membrane performance was highly stable after 240 h of MD operation [93].

## 4. Conclusions and Future Outlook

To achieve improved production cycles inspired by the process intensification (PI) strategy, MD technology is expected to provide a fundamental contribution to seawater desalination. Desalination plants make it possible to ameliorate and preserve natural water resources by releasing water for agriculture and restoring rivers and forests. Nonetheless, the use of membrane distillation in industry, especially in solar energy systems, is particularly limited due mainly to the polarization temperature phenomenon, which reduces the feed temperature at the membrane surface, causing a decrease in the driving force of water evaporation.

In recent years, the integration of plasmonic materials in membrane desalination technology has been demonstrated to be able to significantly improve the thermal efficiency of the MD process through localized thermal heating. At the same time, it could mitigate the lack of effective systems for collecting clean water in the current field of solar steam generation. Furthermore, operating at low pressure, photothermal MD is adapted to treat high-salinity water with a high salt release rate. From the energetic consumption point of view, it is a less energy-intensive new technology compared with conventional MD and RO processes, as it operates mainly by utilizing solar energy. In this scenario, photothermal MD could become one of the most promising technologies for the production of clean water in remote off-grid areas. In addition, the combination of photothermal MD and energy generation may provide a viable solution to the water–energy nexus. Photothermal MD not only supports the production of fresh water by desalination but could also be integrated into systems that allow the production of fresh water and clean energy in parallel. While photothermal MD systems can reach higher thermal efficiencies than conventional MD processes, their thermal efficiencies are lower than those associated with solar steam generation systems [95,98]. Therefore, further efforts should be directed to develop photothermal materials with efficient light-to-heat conversion. For instance, a photothermal layer with micro/nano-rough structures could minimize the phenomena related to reflection and refraction, and improve the absorption of the light. A detailed understanding of the kinetics of light absorption and thermal diffusion of photothermal membranes is also required. More studies are necessary to examine the wetting [98] and fouling of the plasmonic photothermal membranes, problems causing a reduction in the solar conversion efficiency. Furthermore, because most photothermal membranes are made of polymer materials that can degrade under prolonged exposure to sunlight, an analysis of their long-term operation stability becomes necessary. The practical application of photothermal plasmonic membranes is also limited by the possible detachment of the nanoparticles from the membrane substrate during MD photothermal processes. To prevent this, it should be essential to carefully assess the adhesion strength between nanoparticles and the membrane substrate. Regarding the industrial scale-up of the photothermal MD process, it is still at an early stage and requires further progress in membrane module design, particularly in the design of the feed channel. Lastly, an evaluation of costs is required to assess the competitiveness of the photothermal MD technology compared to the other desalination processes.

## Figures and Tables

**Figure 1 membranes-13-00254-f001:**
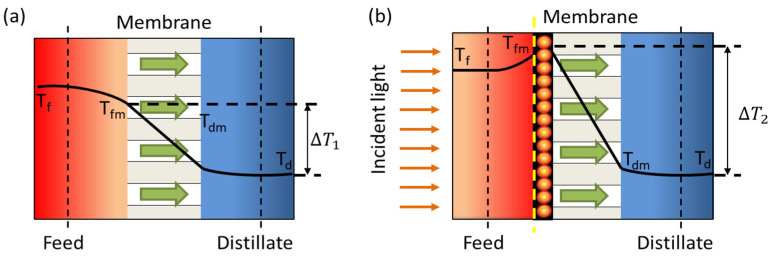
Temperature polarization in (**a**) conventional MD and (**b**) photothermal MD.

**Figure 2 membranes-13-00254-f002:**
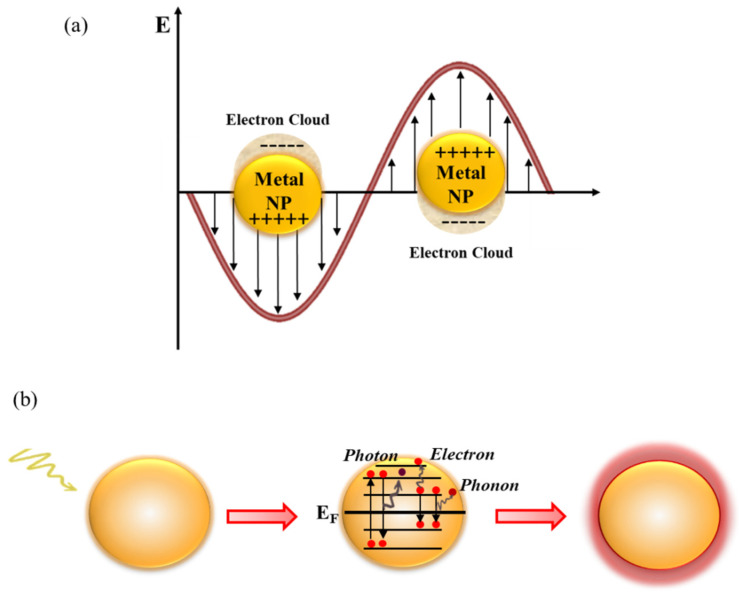
(**a**) Schematic illustration of localized surface plasmon resonance (LSPR). (**b**) Decay channels for plasmonic excitation: electron-to-photon, electron-to-electron, and electron-to-phonon. (NP = plasmonic nanoparticle; E_F_ = Fermi level).

**Figure 3 membranes-13-00254-f003:**
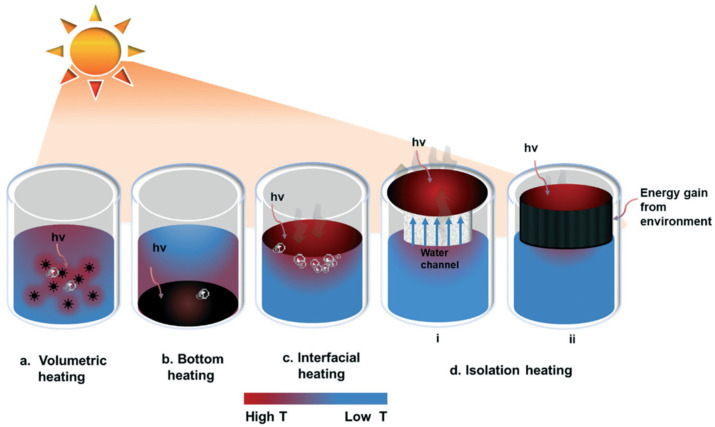
Solar-driven evaporation systems. Reprinted with permission from Ref. [29]. Copyright 2022 Royal Society of Chemistry.

**Figure 4 membranes-13-00254-f004:**
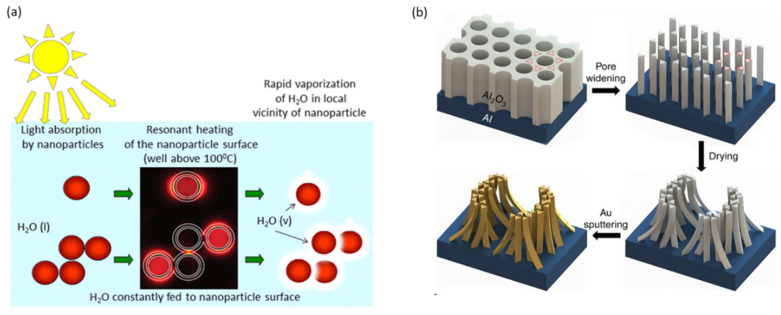
(**a**) SiO_2_/Au nanoparticles and (**b**) schematic illustration for prepare black gold membranes by using pore-widening method into hexagonal anodic aluminum oxide (AAO) templates. (**a**) Reprinted with permission from Ref. [45]. Copyright 2013 American Chemical Society. (**b**) Reprinted from Ref. [53] (open access).

**Figure 5 membranes-13-00254-f005:**
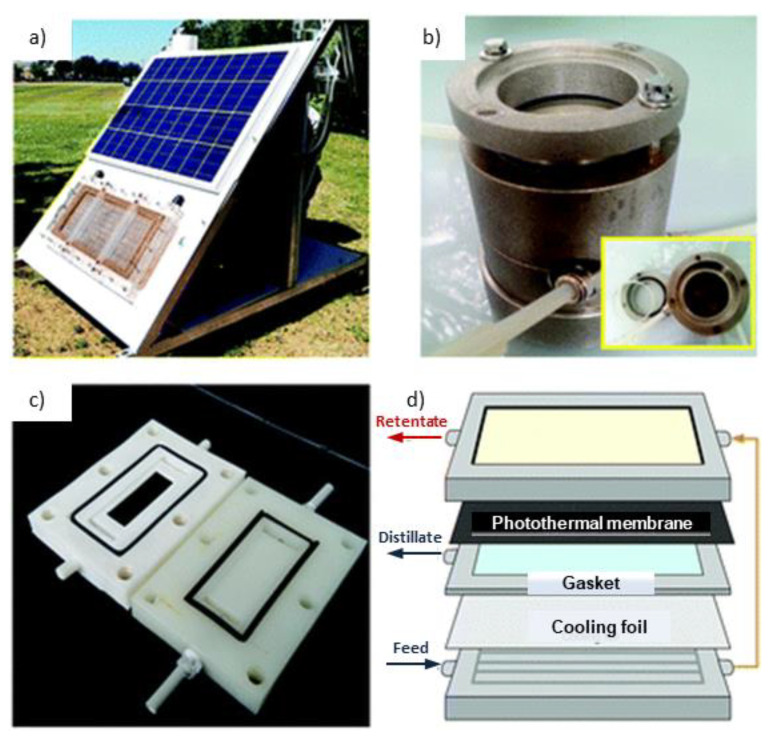
Prototype equipped with a solar panel to provide electric power for the feed and permeate/vacuum pump in PVMD (**a**). Membrane modules employed in lab activities for experiments of PVMD (**b**), PDCMD or PSGMD (**c**). Scheme of a membrane module for PAGMD (**d**). Reproduced with permission from Ref. [43]. Copyright 2022 Royal Society of Chemistry.

**Figure 6 membranes-13-00254-f006:**
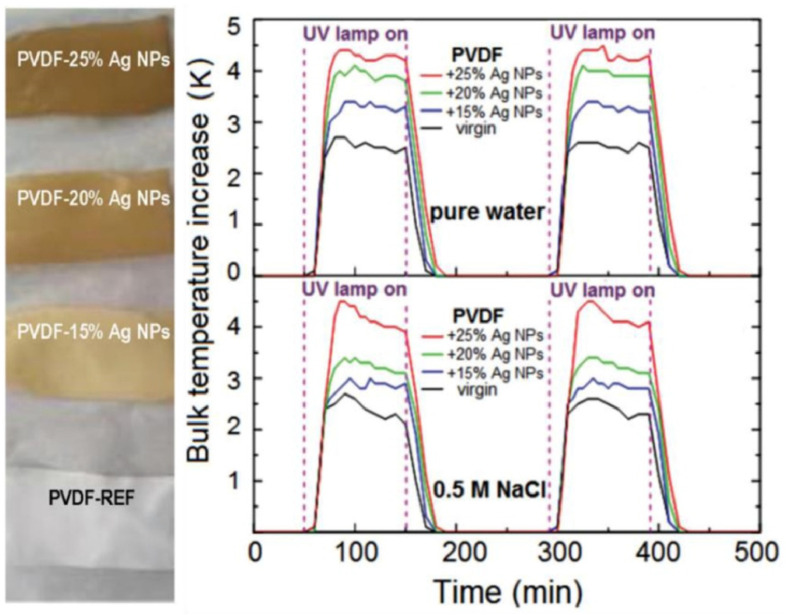
Membranes with different Ag NP loading amounts (**left**) and the effect of Ag NP concentration on the bulk temperature (**right**). Reprinted with permission from Ref. [21]. Copyright 2021 Elsevier.

**Table 1 membranes-13-00254-t001:** Advantages and disadvantages of basic MD configurations.

MD Configuration	Advantages	Disadvantages
DCMD	- Easy scale-up - The permeate is condensed inside the membrane module - Suitable to remove volatile components	- High conductive heat loss - Flux lower than VMD under the same operating conditions
AGMD	- Low conductive heat loss - The permeate is condensed inside the membrane module (integrable with heat recovery) - Less fouling tendency	- Higher mass transfer resistance - Complex module design
SGMD	- Low conductive heat loss	- The permeate is condensed outside the membrane module - Low flux
VMD	- High flux - Very low conductive heat loss	- The permeate is condensed outside the membrane module - Require vacuum pump external condenser - Higher fouling

**Table 2 membranes-13-00254-t002:** Properties, parameters, and performances of plasmonic material-based solar evaporators. Modified from [29]. Reprinted with permission from Ref. [29]. Copyright 2022 Royal Society of Chemistry.

Solar Absorbers	Particle Size (nm)	Pore Size (mm)	Power Density (kW m^−2^)	Wavelength Range (nm)	Absorption (%)	Solar-Thermal Efficiency (%)	Solar-Steam Efficiency (%)	Flux, (kg m^−2^ h^−1^)	Stability/Scalability	Surface Property	Ref.
Black Au-coated Al nanowires	-	≈0.15	20	400–2500	91	57	-	15.95	-	Hydrophilic	[53]
Black Ag NPs	-	-	1	400–1000	-	-	95.2	1.38	10 cycles	-	[63]
Al NPs deposited on AAM	Random	-	4	AM1.5G	96	-	88.4	≈5.7	25 cycles	-	[72]
Cu NPs	≈50	-	2	200–1300	≈97.7	93	73	≈1	Stable/scalable	-	[70]
Ti_3_C_2_ MXene	-	5-35	1	200–2500	96	≈100	87	1.46	-	Hydrophobic	[77]
WO_3−x_ NR-decorated wood	10	-	1	200–1800	94	-	82.5	1.28	-	Hydrophilic	[85]

**Table 3 membranes-13-00254-t003:** Heat transfer at the distillate side for different PMD configurations. Reproduced with permission from Ref. [43]. Copyright 2022 Royal Society of Chemistry.

Configuration	Heat Flux at the Distillate Side	Equation
DCMD-based PMD	Convective heat flux within the distillate channel	$q_{conv}=h_{d}\;\left(T_{m}^{d}-T_{\infty }^{d}\right)$
SGMD-based PMD	Convective heat flux within the coolant channel	$q_{conv}=h_{sweep}\;\left(T_{m}^{g}-T_{\infty }^{g}\right)$
AGMD-based PMD	Conductive heat flux across the air gap	$q_{cond}=\frac{k_{air-vapor}}{\delta_{air\;gap}}\;\left(T_{m}^{a}-T_{satur}\right)$
	Latent heat flux of water vapor across the air gap	$q_{conv}=J\times \Delta H_{v}$
	Conductive heat flux within the condensing film layer	$q=h_{Nusselt}\;\left(T_{satur}-T_{plate}^{d}\right)$ $h_{Nusselt}=0.943\;{\left[\frac{\rho_{liquid\left(\rho_{liquid}-\rho_{vapor}\right)\times \lambda \times g\times k_{liquid}^{3}}}{\mu_{liquid}\times L_{plate}\left(T_{satur}-T_{plate}^{d}\right)}\right]}^{0.25}$
	Conductive heat flux across the cooling plate	$q_{cond}=\frac{k_{plate}}{\delta_{plate}}\;\left(T_{plate}^{d}-T_{plate}^{c}\right)$
	Convective heat flux within the coolant channel	$q_{conv}=h_{coolant}\;\left(T_{plate}^{c}-T_{cool}\right)$

**Table 4 membranes-13-00254-t004:** Examples of plasmonic membranes from the literature. Modified from [43]. Reproduced with permission from Ref. [43]. Copyright 2022 Royal Society of Chemistry.

PMD Configuration	Membrane	Feed	PMD Operative Conditions	Flux (kg m^−2^ h^−1^)	Ref.
DCMD-based PMD	TiN-PVA-PVDF	T_f_ = 23.1 °C NaCl 3.5%	P_in_ = 1kW m^−2^ T^f^_m_ = 39.4 °C (dry) Q_f_ = 25 mL min^−1^	1.0	[91]
DCMD-based PMD	MXenes-PVDF	T_f_ = 65 °C NaCl 1%	P_in_ = 5.8kW m^−2^ T^f^_m_ = 49 °C (dry) Q_f_ = 250 mL min^−1^	8–10	[92]
AGMD-based PMD	TiN-PVA-PVDF	T_f_ = 20 °C NaCl 3.5%	P_in_ = 1kW m^−2^ T^f^_m_ = 58 °C (dry) Q_f_ = 0 mL min^−1^	0.9	[93]
SGMD-based PMD	Ag-PVDF non-woven tissue	T_f_ = 25 °C NaCl 3%	P_in_= 23kW m^−2^ T^f^_m_ = 41 °C (dry) Q_f_ = 50 mL min^−1^	8.6	[94]
VMD-based PMD	Ag-PVDF	T_f_ = 30 °C NaCl 0.5%	P_in_ = 23kW m^−2^ T^f^_m_ = 54 °C (dry) Q_f_ = 330 mL min^−1^	26	[28]

## Data Availability

Not applicable.

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
