# Peer review of "Plasmonic Phenomena in Membrane Distillation"

_membranes, 2023, doi:10.3390/membranes13030254_

Round 1
Reviewer 1 Report
As a review article the manuscript entitled "Plasmonic phenomena in membrane distillation" is very well crafted with enough literature survey and references. However, the first two sections of the article are simply general aspects of the MD process that everyone can read in many other MD-based articles. The practical application of thermoplasmonic membrane distillation is still a matter of concern because it is not only limited to hollowfiber membranes and spiral-wound membrane modules but also limited to multi-modular plate-and-frame modules because of the limited membrane exposure to solar radiation in the inner stages of the membrane module. Hopefully, future research would find a way to overcome the aforementioned practical difficulties. I recommend this review article to publish in the "Membranes" journal as it is.
Author Response
We thank the reviewer for the positive comments. As suggested, the first two sections of the article have been delated and the main aspects incorporated into the Introduction. The disadvantages, challenges and prospects are also described.
Reviewer 2 Report
This manuscript reviews the use of plasmonic nanostructures to reduce or eliminate temperature polarization which in turn to improve the MD flux. This is an interesting subject in MD technology. However, a few of recommendations/comments which should be considered before acceptance for publication in this journal.
Abstract:
Line 18: It is confusing for the phrase ‘conversion of heat into light’.
Introduction:
Line 64-69: More explanation on plasmonic phenomena is required. Briefly describe how the plasmonic nanomaterials have improved the MD performance in the previous studies and also compare with the non-plasmonic nanostructures.
Line 73-270: The review on the subject ‘Membrane Distillation’ is quite general. Many review articles on this subject have been published since 1997. Suggesting to shorten these paragraphs and/or may be appropriate to combine into the Introduction section.
Line 293-295: Should be excluded.
Line 271-400: This section introduces thermoplasmonics concept. However, it will be better if this concept incorporates with MD. In my opinion, only the equations that related to MD should be explained here. If explaining the general concept of thermoplasmonics only, author may consider to combine in the Introduction section.
Line 403-484: Author may consider to present some of the information in table forms as well as graphics/figures/diagrams. A table can be used to describe and compare for different parameters/variables such as plasmonic volumetric system/interfacial system, types of NPs, evaporation rate/performance, photothermal conversion, design, radiation time, etc. Suggesting some relevant equations as described in section 3 can be included here.
Line 511-537: Citation of references is required. Author may consider to include a table in section 5, especially the fabrication of plasmonic nanostructure membranes for MD and how much of the improvement.
Line 608-622: Schematic diagrams for various module designs are encouraged.
Author may consider to combine section 7 into section 5.
References:
Almost 50% of the literature cited was more than 5 years. Suggesting to include more recent papers relating to this subject.
Author Response
Reviewer 2:
This manuscript reviews the use of plasmonic nanostructures to reduce or eliminate temperature polarization which in turn to improve the MD flux. This is an interesting subject in MD technology. However, a few of recommendations/comments which should be considered before acceptance for publication in this journal.
Abstract:
Line 18: It is confusing for the phrase ‘conversion of heat into light’.
Authors’ answer: The phrase ‘conversion of heat into light’ has been corrected as follows: ‘conversion of light into heat’.
Introduction:
Line 64-69: More explanation on plasmonic phenomena is required. Briefly describe how the plasmonic nanomaterials have improved the MD performance in the previous studies and also compare with the non-plasmonic nanostructures.
Authors’ answer: The Introduction section has been modified, sections 2 and 3 of the original version of the manuscript deleted and the main aspects incorporated in the Introduction. The plasmonic heating mechanism has been described in the Introduction section, the plasmonic effect on MD process is illustrated in the new and revised version of section 3 “Photothermal membrane distillation”.
The performance of MD when plasmonic materials are used with respect to MD process utilizing membranes without plasmonic materials can be found in Introduction as follow:
“However, the traditional MD process is characterized by considerable drawbacks, including high energy input for heating the bulk feed water, heat loss during the feed transportation from heating units to membrane modules, and the need for large central pumping systems. In addition, its large-scale application is also limited by its low thermal efficiency due to the temperature polarization (TP). This phenomenon is related to latent heat and conductive heat transfer at the MD membrane interface. The feed temperature at the membrane surface is lower than the bulk feed water temperature due to thermal conductivity and water vaporization. Likewise, the temperature at the distillate-membrane interface is higher than the temperature of the bulk distillate . Therefore, TP reduces significantly the temperature difference at the membrane interface with respect to the theoretical driving force across the bulks (Figure 1a).
Several solutions have been suggested to mitigate the negative effects of TP. Some studies reported the use of frame-like turbulence promoters (e.g., feed spacers) or modified feed channels (e.g., corrugated channels) to alleviate TP and, thus, control MD performance and improve distillation efficiency. Nevertheless, the use of these approaches leads to an increased demand for energy.
Because TP is an inherent loss process that cannot be completely removed, it is desirable to also explore other strategies that could alleviate this phenomenon and improve MD performance.
In particular, thermoplasmonics could offer new opportunities for MD, thanks to the possibility of harnessing the kinetic energy of the light to generate and control a large amount of heat at the nanoscale. Over the past few years, the use of photothermal materials acting as nanosources of heat in the MD membranes has been proposed for TP mitigation, contributing to the development of a new MD configuration, called photothermal membrane distillation (PMD).
In the literature, various photothermal materials developed for solar steam generation (SSG) systems have been employed to realize PMD membranes, including plasmonic materials [], carbon-based nanomaterials [], inorganic semiconductors [], and polymers [].
Most notably, plasmonic nanostructures (Au, Ag, MXene, TiN, etc.) have intensive light absorption and strong photothermal conversion capacities.
The plasmonic heating mechanism is based on a nonradiative conversion of light to heat by plasmonic nanostructures. As shown in Figure 2a, plasmonic materials exhibit localized surface plasmon resonance (LSPR) when the frequency of the incident light matches the resonance frequency of the free electrons. The excitation of electrons is triggered by the match between the frequencies. Following the photoexcitation, the surface plasmons can behave as strongly damped oscillators and decay through various channels of interaction [53] (electron-to-photon, electron-to-electron, and electron-to-phonon) that allow the dissipation of the plasmonic energy in thermal energy (Figure 2b).
Specifically, in the timescale ranging from 1 to 100 after Landau damping, the electron-hole pairs can decay either by emission of photon or by carrier multiplication due to the electron-electron interactions [53,54]. The hot carriers will redistribute their energy via electron-electron scattering processes in a few hundred femtoseconds. Lastly, the heat will dissipate by thermal conduction in the environment surrounding the nanostructure on a time scale between 100 ps and 10 ns.
For a single metallic NP surrounded by a homogeneous medium, the increase in temperature due to the light irradiation is given by
where is the absorption cross-section, is the irradiance of the illumination, is the radius of the NP, and is the thermal conductivity of the surrounding medium.
In general, the contribution of a sequence of metallic NPs surrounded by a homogeneous medium can be obtained by solving the following heat transfer equation
where, , , are the density, specific heat capacity, temperature, thermal conductivity and local heat intensity, respectively, expressed as a function of coordinate .
As a result, it is difficult to assess the impacts of nanosources of heat on the membrane temperature.
Recently, Elmaghraoui et al. [] modified the previous model by the addition of a term associated with heat losses into the heat transfer equation. This new approach includes the cooling effects related to vaporization at the membrane surface. According to this, the membrane steady-state temperature is reported in the following equation
where is the effectively continuous heat production due to absorbed energy on NPs that depends on the excitation energy, the intensity, the NP concentration and their size, is a radium of a sphere with a volume of measured suspensions, and is a cooling time constant. The introduction of the cooling term in the theoretical model, provides an even more accurate description of the experimental results.
Both models suggest that the membrane temperature increases with the number density of the NPs used as filler. Therefore, the use of nanosources of heat in MD processes would lead to a temperature increase in a large area (Figure 1b), resulting in an improvement in mass transfer.
However, to date, the practical application of photothermal membranes is hindered by many obstacles including membrane wetting, fouling, scaling degradation and last but not least the need to redesign MD membrane module.”
The performance of MD when plasmonic materials are used with respect to MD process utilizing membranes without plasmonic materials can be also found in the new section 3.4. In particular, in Section 3.4, both the interface temperature and the trans-membrane flux of virgin (and unloaded ) PVDF membranes have been compared with those of PVDF membrane loaded with Ag NPs.
Line 73-270: The review on the subject ‘Membrane Distillation’ is quite general. Many review articles on this subject have been published since 1997. Suggesting to shorten these paragraphs and/or may be appropriate to combine into the Introduction section.
Authors’ answer: Section 2 of the original version of the manuscript has been eliminated and the main aspects of Membrane Distillation have been incorporated into the new version of the Introduction section.
Line 293-295: Should be excluded.
Authors’ answer: Lines 293-295 have been excluded.
Line 271-400: This section introduces thermoplasmonics concept. However, it will be better if this concept incorporates with MD. In my opinion, only the equations that related to MD should be explained here. If explaining the general concept of thermoplasmonics only, author may consider to combine in the Introduction section.
Authors’ answer: As well as Section 2, also Section 3 of the original version of the manuscript has been eliminated and the main aspects have been incorporated into the Introduction section.
Line 403-484: Author may consider to present some of the information in table forms as well as graphics/figures/diagrams. A table can be used to describe and compare for different parameters/variables such as plasmonic volumetric system/interfacial system, types of NPs, evaporation rate/performance, photothermal conversion, design, radiation time, etc. Suggesting some relevant equations as described in section 3 can be included here.
Authors’ answer: The manuscript has been modified, Section 2 and 3 of the original version deleted, only the main equations describing the phenomena reported in the article. Various new figures and tables have been added in the revised version of the manuscript (such as Table 2 and 3, Figure 1, 3 and 5).
Line 511-537: Citation of references is required. Author may consider to include a table in section 5, especially the fabrication of plasmonic nanostructure membranes for MD and how much of the improvement.
Authors’ answer: References have been corrected and up-dated. In the revised version of the manuscript, the original section 5 became section 3, where “Table 3. Examples of plasmonic membranes from literature” has been added.
Line 608-622: Schematic diagrams for various module designs are encouraged.
Authors’ answer: Figure 4 with pictures or scheme of membrane modules utilized in experiments of PVMD, PDCMD or PSGMD, PAGMD has been added.
Author may consider to combine section 7 into section 5.
Authors’ answer: Done. Section 7 of the original version of the manuscript has been incorporated into the new version of Section 3 (i.e., Section 5 of the original version of the manuscript).
References:
Almost 50% of the literature cited was more than 5 years. Suggesting to include more recent papers relating to this subject.
Authors’ answer: The manuscript has been revised, sections 2 and 3 deleted, description of plasmonic phenomena and materials added. Therefore, reference list of the revised version of the manuscript has been up-dated and 52% of the cited manuscripts are from 2018.
Reviewer 3 Report
The authors presented a review article on the possibility of using Plasmonic phenomena in the process of membrane distillation. However, this work requires a decisive re-editing, as well as the removal of many pages of a well-known description of MD, which in its content is not related to the topic of Plasmonic phenomena.
Plasmonic phenomena research is interesting from the point of view of academic research, but the application possibilities are doubtful - this should be clearly emphasized to the reader, rather than creating the impression of an emerging solution that will overcome the problems of MD.
To determine the possibility of using solar energy, two basic pieces of information should be taken into account:
1) It takes 2500 kJ/kg to evaporate the water
2) On Earth, sunlight is scattered and filtered through Earth's atmosphere, and is obvious as daylight when the Sun is above the horizon. When direct solar radiation is not blocked by clouds, it is experienced as sunshine, a combination of bright light and radiant heat. When blocked by clouds or reflected off other objects, sunlight is diffused. Sources estimate a global average of between 164 watts to 340 watts per square meter over a 24-hour day (NASA data).
For these reasons, it is difficult to build large solar installations - rather, they will practically be small home MDs. Hence, the Introduction should be improved, showing real possibilities. Sentences like the following are not this:
“Recently, plasmonic nanostructures are emerging as one of the most exciting candidates for solar steam generation [19,20].”
In addition, many studies, as in photocatalysis, clearly indicate that polymeric membranes cannot be irradiated. Hence such a valid suggestion, however, introduces unfounded hope:
“Because most photothermal membranes are made of polymer materials that can degrade under prolonged exposure to sunlight, an analysis of their long-term operation stability becomes necessary.”
With current documented knowledge, we know that such an application is impossible for polymeric membranes.
Detailed notes:
1) Plasmonic phenomena works independently of the MD variant, so why distract the reader with such information, hence remove section 2 - only give a general drawing of the MD principle with the phenomenon of polarization.
2) What are the descriptions of the membrane material supposed to give? – it would make sense to discuss how the properties of membranes change due to the addition of structures that cause plasmonic phenomena. Similarly for MD equations - what would it change in them? - and yes, it's just unrelated ballast.
3) Section 3
a) in addition to metallic, this effect is also shown by non-metallic nanostructures - much cheaper than the use of Au.
b) Efficiency is not indicated, what percentage of light energy is converted into heat.
c) We do not desalinate distilled water - there is nothing about the effects of fouling or scaling “To give an order of magnitude, a spherical Au NP with a diameter of 20 ??, illuminated at ? = 530 ?? with an irradiance of ?=1 ??/??2 experiences a temperature increase of ∼ 5 °?.”
So it is 10E9 W/m2 - a value impossible to obtain for solar irradiation.
4) “This film revealed a solar thermal conversion efficiency of up to 57% at 20 kW/m2.”
- where on Earth does 20 kW/m2 come from?!
Authors should critically analyze published information. Experience should be distinguished when e.g. 2x2 cm membrane is illuminated with a 100W bulb,
5) Fig.5
- if we cover the membranes with such protruding structures, we greatly increase laminar layer mixing, which reduces polarization. We know for many decades that such structures on the surface even at low Pe values, e.g. 50, they can cause a turbulent flow.
- did the work showing the increase in efficiency take this into account - maybe this was the reason for the increase in permeate flux?
6) „Copper is envisioned as a promising alternative to Au and Ag as it costs significantly less. Zhang et al. [84]”
The Cu layer will dissolve under the influence of seawater - information from other publications should be given/evaluated critically.
7) Fig. 6b
- after starting the additional heat source, an equilibrium state will be created - then at most TF=TFm - the feed in the bulk will also be heated.
8) Fig. 7
Flux 10e3 kg/m2s??? rather 10e-3 kg/m2s - which is still suspiciously high efficiency.
9) “This temperature difference could be further increased by reducing conductive heat loss from the feed side to the permeated side.”
- we increase the value of TFm, hence the losses will be even greater because there is no polarization.
10) Section 8. Conclusions and future outlook
It should demonstrate the more realistic possibilities of plasmonic materials, more critically assess the previous experience.
Author Response
Reviewer 3:
The authors presented a review article on the possibility of using Plasmonic phenomena in the process of membrane distillation. However, this work requires a decisive re-editing, as well as the removal of many pages of a well-known description of MD, which in its content is not related to the topic of Plasmonic phenomena.
Plasmonic phenomena research is interesting from the point of view of academic research, but the application possibilities are doubtful - this should be clearly emphasized to the reader, rather than creating the impression of an emerging solution that will overcome the problems of MD.
To determine the possibility of using solar energy, two basic pieces of information should be taken into account:
1) It takes 2500 kJ/kg to evaporate the water
2) On Earth, sunlight is scattered and filtered through Earth's atmosphere, and is obvious as daylight when the Sun is above the horizon. When direct solar radiation is not blocked by clouds, it is experienced as sunshine, a combination of bright light and radiant heat. When blocked by clouds or reflected off other objects, sunlight is diffused. Sources estimate a global average of between 164 watts to 340 watts per square meter over a 24-hour day (NASA data).
For these reasons, it is difficult to build large solar installations - rather, they will practically be small home MDs. Hence, the Introduction should be improved, showing real possibilities. Sentences like the following are not this:
“Recently, plasmonic nanostructures are emerging as one of the most exciting candidates for solar steam generation [19,20].”
“Recently, plasmonic nanostructures are emerging as one of the most exciting candidates for solar steam generation [19,20].”
In addition, many studies, as in photocatalysis, clearly indicate that polymeric membranes cannot be irradiated. Hence such a valid suggestion, however, introduces unfounded hope:
“Because most photothermal membranes are made of polymer materials that can degrade under prolonged exposure to sunlight, an analysis of their long-term operation stability becomes necessary.”
With current documented knowledge, we know that such an application is impossible for polymeric membranes.
Authors’ answer: The manuscript has been completely revised accordingly to the three reviewers’ comments. In particular, sections 2 and 3 of the original version of the manuscript have been delated and the main concepts incorporated in the Introduction section. The tone of various sentences mitigated. Moreover, in addition to the possible and potential advantages, also the disadvantages and challenges have been described along the text indicating that more studies are necessary.
Examples of the changes are as follows:
-“However, to date, the practical application of photothermal membranes is hindered by many obstacles including membrane wetting, fouling, scaling degradation and last but not least the need to redesign MD membrane module.”
- “In spite of the photothermal MD systems can reach higher thermal efficiencies than conventional MD processes, their thermal efficiencies are lower than those associated with solar steam generation systems [95,98]. Therefore, further efforts should be directed to develop photothermal materials with efficient light-to-heat conversion. For instance, a photothermal layer with micro/nano-rough structures could minimize the phenomena related to reflection and refraction, and improve the absorption of the light. A detailed under-standing of the kinetics of light absorption and thermal diffusion of photothermal mem-branes is also required. More studies are necessary to examine the wetting [98] and fouling of the plasmonic photothermal membranes, problems causing able to a reduction of the solar conversion efficiency. Also, because most photothermal membranes are made of polymer materials that can degrade under prolonged exposure to sunlight, an analysis of their long-term operation stability becomes necessary. The practical application of photo-thermal plasmonic membranes is also limited by the possible detachment of the nanoparticles from the membrane substrate during MD photothermal processes. To prevent this, should be essential to carefully assess the adhesion strength between nanoparticles and the membrane substrate. Regarding the industrial scale-up of the photothermal MD process, it is still at an early stage and requires further progress in membrane module design, in particular in the design of the feed channel. Lastly, an evaluation of costs is required to assess the competitiveness of the photothermal MD technology compared to the other desalination processes.”
Detailed notes:
- Plasmonic phenomena works independently of the MD variant, so why distract the reader with such information, hence remove section 2 - only give a general drawing of the MD principle with the phenomenon of polarization.
Authors’ answer: Section 2 has been removed in the revisited manuscript. General drawing of the MD principle with the temperature polarization is presented in the Introduction section.
- What are the descriptions of the membrane material supposed to give? – it would make sense to discuss how the properties of membranes change due to the addition of structures that cause plasmonic phenomena. Similarly for MD equations - what would it change in them? - and yes, it's just unrelated ballast.
Authors’ answer: Section 2.3 of the original version of the manuscript has been deleted whereas more information on the different materials with plasmonic photothermal properties and examples of plasmonic membranes are described in section 3.4 of the revised version of the manuscript.
3) Section 3
- a) in addition to metallic, this effect is also shown by non-metallic nanostructures - much cheaper than the use of Au.
Authors’ answer: Topic of the present manuscript is to illustrate the “Plasmonic phenomena in Membrane Distillation”. Therefore, only materials with plasmonic photothermal properties have been considered. Other materials (such as carbon-based and polymer-based materials) presents photothermal properties due to the molecule thermal vibration mechanism. On the contrary, aim of the present manuscript is to describe only the materials with photothermal properties due to the plasmonic localized heating mechanism. Various other reviews already present in literature describe and compare the various plasma, polymer-based and inorganic semiconductor materials.
- b) Efficiency is not indicated, what percentage of light energy is converted into heat.
Authors’ answer: In the revised version of the manuscript, solar-steam and solar-thermal efficiencies are reported in Section 2 and 3, and in Table 2.
- c) We do not desalinate distilled water - there is nothing about the effects of fouling or scaling “To give an order of magnitude, a spherical Au NP with a diameter of 20 ??, illuminated at ?= 530 ??with an irradiance of ?=1 ??/??2 experiences a temperature increase of ∼ 5 °?.”
So it is 10E9 W/m2 - a value impossible to obtain for solar irradiation.
Authors’ answer: The cited sentence has been deleted.
4) “This film revealed a solar thermal conversion efficiency of up to 57% at 20 kW/m2.”
- where on Earth does 20 kW/m2 come from?!
Authors should critically analyze published information. Experience should be distinguished when e.g. 2x2 cm membrane is illuminated with a 100W bulb,
Authors’ answer: The sentence has been corrected as follows: “This Au film recorded 57% conversion efficiency under light illumination of at 20 kW m-2”.
5) Fig.5
- if we cover the membranes with such protruding structures, we greatly increase laminar layer mixing, which reduces polarization. We know for many decades that such structures on the surface even at low Pe values, e.g. 50, they can cause a turbulent flow.
- did the work showing the increase in efficiency take this into account - maybe this was the reason for the increase in permeate flux?
Authors’ answer: Figure 5 of the original version of the manuscript is Figure 4 in the revised version. The work of Bae et al. [53] has been reported only as an example of multiscale structures for enhancing light absorption and low reflectance. We never cited the trans-membrane flux achieved by Bae et al., neither in the original version nor in the revised version of the manuscript. Surely, the increase of membrane roughness improves MD performance.
6) „Copper is envisioned as a promising alternative to Au and Ag as it costs significantly less. Zhang et al. [84]”
The Cu layer will dissolve under the influence of seawater - information from other publications should be given/evaluated critically.
Authors’ answer: The text has been modified as follows:
“A cheaper alternative to Au and Ag may be represented by copper (Cu) and aluminium (Al). Due to its strong vis-NIR absorption and omnidirectional light absorption properties, Cu nanoparticles show LSPR peaks at longer wavelengths compared to AuNPs and AgNPs [66]. Nevertheless, Cu NPs exhibit less stability to corrosion in aqueous solutions and oxidation in air than Au and Ag NPs. To overcome these obstacles, Cu NP films have been covered with thin protective coatings or treated with corrosion inhibitors [67,68]“.
7) Fig. 6b
- after starting the additional heat source, an equilibrium state will be created - then at most TF=TFm - the feed in the bulk will also be heated.
Authors’ answer: Figure 6b of the original version of the manuscript has been modified and the new version is named Figure 1 in the revised version of the manuscript. In the new version, the temperature increase of the feed at membrane surface and in the bulk in PMD compared to conventional MD is clearly drawn.
8) Fig. 7
Flux 10e3 kg/m2s??? rather 10e-3 kg/m2s - which is still suspiciously high efficiency.
Authors’ answer: We thank the reviewer for the comment. We correct the mistake and the up-dated version of the figure can be found as Fig. 5.
9) “This temperature difference could be further increased by reducing conductive heat loss from the feed side to the permeated side.”
- we increase the value of TFm, hence the losses will be even greater because there is no polarization.
Authors’ answer: The localized heating by the photothermal active layer at the evaporation surface leads to an increase in feed temperature at the membrane surface, and thus in the driving force ΔTm. Therefore, heat loss from the feed to the distillate side through conduction should be minimized. Thermal conductivity of the membrane plays an important role in the heat transfer resistance between the feed and the distillate. In general, the low thermal conductivity of the membrane is required to prevent heat loss to the distillate side, this can be achieved by increasing porosity and thickness of the membrane, or reducing the thermal conductivity of the membrane material.
The text in section 3 has been modified as follows:
“Localized heating causes an increase in the feed temperature at the membrane surface, resulting in an increase in the temperature difference between the feed and the distillate sides and thus in the driving force of the process. This temperature difference could be further increased by reducing conductive heat loss from the feed side to the permeated side. A low thermal conductivity of the membrane material, together with high membrane porosity and thickness, are key factors for minimizing the heat transfer between feed and permeate [95].”
10) Section 8. Conclusions and future outlook
It should demonstrate the more realistic possibilities of plasmonic materials, more critically assess the previous experience.
Authors’ answer: Disadvantages and challenges of PMD have been described along the text in various sections as well as in the “Conclusions and future outlook” as follows:
“While photothermal MD systems can reach higher thermal efficiencies than conventional MD processes, their thermal efficiencies are lower than those associated with solar steam generation systems [95,98]. Therefore, further efforts should be directed to develop photothermal materials with efficient light-to-heat conversion. For instance, a photothermal layer with micro/nano-rough structures could minimize the phenomena related to reflection and refraction, and improve the absorption of the light. A detailed understanding of the kinetics of light absorption and thermal diffusion of photothermal membranes is also required. More studies are necessary to examine the wetting [98] and fouling of the plasmonic photothermal membranes, problems causing a reduction of the solar conversion efficiency. Also, because most photothermal membranes are made of polymer materials that can degrade under prolonged exposure to sunlight, an analysis of their long-term operation stability becomes necessary. The practical application of photothermal plasmonic membranes is also limited by the possible detachment of the nanoparticles from the membrane substrate during MD photothermal processes. To prevent this, it should be essential to carefully assess the adhesion strength between nanoparticles and the membrane substrate. Regarding the industrial scale-up of the photothermal MD process, it is still at an early stage and requires further progress in membrane module design, in particular in the design of the feed channel. Lastly, an evaluation of costs is required to assess the competitiveness of the photothermal MD technology compared to the other desalination processes.”
Round 2
Reviewer 2 Report
This is a revised version of the manuscript that reviews the use of plasmonic nanostructures to reduce or eliminate temperature polarization which in turn to improve the MD flux. The author has majorly improved and revised the manuscript. Therefore, I agree to accept the present form of the manuscript for publication in this journal.